# Consumer Preferences, Sensory Evaluation, and Color Analysis of Beetroot and Tomato Juices: Implications for Product Development and Marketing in Health-Promoting Beverages

**DOI:** 10.3390/foods13244059

**Published:** 2024-12-16

**Authors:** Marek Kardas, Michalina Rakuła, Aleksandra Kołodziejczyk, Wiktoria Staśkiewicz-Bartecka

**Affiliations:** Department of Food Technology and Quality Assessment, School of Public Health in Bytom, Medical University of Silesia in Katowice, ul. Jordana 19, 41-808 Zabrze, Poland; mkardas@sum.edu.pl (M.K.); mrakula@sum.edy.pl (M.R.); olap12141990@gmail.com (A.K.)

**Keywords:** health-promoting foods, consumer preferences, CIELAB color system, sensory analysis

## Abstract

Background/Objectives: This study explores the significance of beetroot and tomato juices, two prominent health-promoting foods known for their rich nutrient content and bioactive compounds. The growing consumer awareness of the link between diet and well-being emphasizes the need for food producers to align their products with health-conscious preferences. The aim of this research was to assess the composition, color, and sensory attributes—specifically color, taste, and odor—of various commercially available beetroot and tomato juices and to evaluate their acceptability among consumers. Methods: A total of 50 dietitians (41 women and 9 men) participated in sensory evaluations and spectrophotometric color analysis using the CIELAB system, which measures lightness (L*), red–green tones (a*), and blue–yellow tones (b*). This dual approach allowed for a comprehensive understanding of how color characteristics correspond to sensory ratings. Results: Results revealed significant differences in color and sensory attributes among the juices, with darker hues and higher red-tone values generally preferred by consumers. Juices with lower lightness (L*) and dominant blue or red tones (negative b*, higher a*) were consistently rated higher, suggesting that color plays a pivotal role in initial product acceptance. However, no single juice excelled across all sensory categories, indicating varied consumer preference. Conclusions: The findings underscore the influence of color on consumer perception and its potential for guiding product development. For producers of functional beverages, optimizing visual appeal through precise control of color parameters could enhance marketability while balancing sensory attributes such as taste and aroma. These insights support the development of products that satisfy both nutritional goals and consumer expectations.

## 1. Introduction

The role of food in enhancing well-being and mitigating the risk of chronic disease is significant. This has led to an increased awareness among consumers of the impact of diet on health, which should encourage manufacturers to enhance the health-promoting attributes of their products [1,2,3,4]. Despite the absence of a unified definition of health-promoting foods, the majority of experts concur that these foods offer proven health and preventive benefits in addition to their nutritional function [5,6,7]. In Poland, the category of health-promoting foods is dominated by dairy products and juices enriched with vitamins and minerals [8,9]. In accordance with dietary recommendations that advocate the consumption of fruit and vegetables as the cornerstone of a healthy diet, fruit and vegetable juices are experiencing a surge in popularity due to their health-promoting attributes and convenient mode of administration [10,11,12,13]. The consumption of fruit and vegetable juices represents a viable means of providing the body with an augmented dose of essential vitamins and minerals and antioxidants. The content of these components is contingent upon the type of raw material, thermal processing methods, and degree of processing and in turn affects their efficacy in the prevention of lifestyle diseases [14,15,16,17].

The utilization of raw materials that are rich in biologically active compounds and devoid of anti-nutritional factors, in conjunction with suitable processing techniques, enables the production of health-promoting products such as beetroot juice [17,18,19,20,21,22,23]. Beetroot is esteemed for its high nutritional value, appealing taste, intense color, and minimal health risks, making it a valuable commodity for both consumers and the food industry. The processing of beetroot, including the production of beetroot juice, is becoming increasingly popular. The variety of beetroot juices available on the market ranges from pure juices with an intense, earthy flavor to blended juices, such as a combination with apple juice, which are milder and more approachable [24,25].

Tomato preparations, such as juices, are valued for their ability to provide key nutrients in a form analogous to raw tomatoes, although the exact amount may vary slightly. Furthermore, processing can enhance the bioavailability of specific bioactive compounds, such as lycopene, a carotenoid with established health benefits [26,27,28,29]. The market for tomato juice is expanding, with an increasing variety of products on offer that differ in terms of taste, production method, and type of packaging. Tomato juices are available in a range of forms, including pure, blended, and with additives such as salt, spices, pepper, chili, or herbs, allowing consumers to select a product that aligns with their preferences. Additionally, combinations of tomato juices with other vegetables are also available [30].

The nutritional value and sensory attributes of food products are significant factors influencing consumer satisfaction, which in turn affects future consumption choices, product development, and quality control [23,31,32]. The acceptability of food products is largely determined by three key factors: color, taste, and texture. Of these, color is arguably the most significant, influencing product evaluation to a considerable extent [23,32,33]. It is of the utmost importance to gain an understanding of consumer needs in order to ensure the production of products that are appropriately adapted to meet those needs. Consumer sensory surveys that quantify the intensity of different sensory attributes using intensity scales can be a valuable source of information for food producers [13].

Color is a significant parameter of food quality, exerting a considerable influence on consumer acceptance and purchase decisions. Nutritionists may consider this aspect when recommending products to their patients, thereby making the understanding of their preferences relevant to food manufacturers [34,35,36]. Furthermore, color can be accurately determined using the CIELAB color space, which defines brightness (L), green–red tones (a), and blue–yellow tones (b) [34,35,37]. The intuitive nature of the CIELAB coordinates facilitates the analysis and presentation of data in the context of scientific research on the colors of food products, such as fruit juices [35].

Color measurements and sensory perception are crucial for product development in the health-food sector, particularly for functional beverages. Recent research highlights the psychological impact of food color on flavor perception, where darker and more saturated hues are often associated with richer flavors and higher nutrient density. Furthermore, consumer studies highlight the importance of aligning visual attributes with health-related expectations. For instance, darker tomato juices with higher red intensity are perceived as more natural and nutritious, while brighter, yellow-tinged juices are often considered less appealing. These insights underscore the necessity of studies with sensory evaluation to create products that not only meet aesthetic standards but also align with consumer expectations of healthfulness and taste. This theoretical framework forms the basis for this study, which aims to provide actionable insights for functional food producers by bridging the gap between instrumental data and consumer behavior [36,38,39].

The aim of this study was to comprehensively examine and characterize a selected category of health-promoting food, in which special attention was paid to beetroot and tomato juices. The study included an assessment of the composition of these products and an analysis of key quality features, such as color, odor, and taste, taking into account consumer preferences. In addition, using the spectrophotometric method in the CIELAB color system, the color of the juices tested was measured, which allowed for a precise comparison of their visual properties. The study also aimed to determine whether there are statistically significant differences in the assessment of individual organoleptic features of these products.

Research hypotheses were formulated to guide the analysis and structure the interpretation of the study’s results. It was hypothesized that the intensity of color and specific colorimetric parameters, measured within the CIELAB system, significantly influence consumer acceptability of beetroot and tomato juices. Additionally, it was assumed that a significant relationship exists between sensory attributes and consumer preferences.

## 2. Materials and Methods

### 2.1. Study Design

A total of 50 dietitians participated in the study, comprising 41 women and 9 men. The age of the respondents was 24 ± 3.21 years. All the students were actively working as dietitians (the students had completed their first degree, qualifying them to practice as dietitians). The study consisted of several stages, conducted from December 2023 to April 2024. The first stage consisted of a review of the assortment of stationary and online stores, and then the selection of specific products for further research. In this part, the assortment was also characterized and the composition of selected products was assessed. The second stage included a survey on preferences, while the third stage included a sensory analysis using the ranking method. The final stage of the study involved the measurement of juice color using the Tri-Color SF80 spectrophotometer (Narama, Poland). The precise structure of the study is presented in Figure 1.

This study was carried out in a sensory analysis laboratory designed according to the PN-EN ISO 8589:2010 standard [40]. The Declaration of Helsinki of the World Medical Association guided the conduct of this study. The study protocol (KNW 0022/KB1/73/I/16) was reviewed by the Bioethics Committee of the Silesian Medical University in Katowice and was approved. Each person participating in the study gave informed consent to participate in the study and was informed about the anonymity of the results.

### 2.2. Research Material

The research material consisted of 6 beetroot juices and 8 tomato juices from different producers (Figure 2).

The products were procured from hypermarkets in Zabrze in January 2024. The juices were stored according to the manufacturer’s recommendations until the tests were conducted. The data provided by the manufacturer on the labels of individual packages were used to evaluate the composition of the analyzed products. Table 1 presents the characteristics of the tested juices.

### 2.3. Research Tools

#### 2.3.1. Preference and Frequency of Consumption

The assessment of preferences and frequency of juice consumption was assessed using an original survey questionnaire, which consisted of a personal data sheet and 6 closed questions. The personal data questions included information such as gender, age, and place of residence, while the remaining questions proper concerned the frequency of juice consumption, preferences for types of juices in terms of raw material and producer, reasons for consuming juices, and factors determining the choice.

#### 2.3.2. Sensory Analysis

The study was conducted using the authors’ questionnaire, employing a serialization method that adheres to the PN-ISO 5497:1998 standard [41].

The analysis was conducted using six beetroot juice samples and eight tomato juice samples, which were randomly coded with three-digit codes to ensure anonymity. The 30 mL samples were presented in homogeneous, transparent plastic cups, having been shaken beforehand. The juices were served at a consistent temperature of 8 °C, which is typical for chilled juice consumption. The samples were presented in a randomized order to avoid any bias related to the sequence of tastings. Each participant was given a short break with water provided to cleanse their palate between tasting different samples, ensuring that the flavor of one sample did not affect the perception of the subsequent ones.

The sensory tests took place in a dedicated sensory analysis laboratory equipped with individual booths to minimize distractions and ensure a controlled environment. Each booth was adequately lit, and the ambient temperature was maintained at 22 °C.

The participants evaluated the samples based on three criteria: color, odor, and taste. The evaluation commenced with beetroot juice and subsequently proceeded to tomato juice. Each set was accompanied by a disposable cup of water, which was provided to neutralize the taste.

The sensory testing was conducted in a dedicated laboratory with separate stations to ensure controlled testing conditions. The investigators’ task was to rank the received juice samples in order from most to least desirable, in terms of color, aroma, and taste, and place the results on the authors’ evaluation sheets. The order in which the products were ranked was assigned ranks. Accordingly, the best sample was given a rank of 1, the next was given a rank of 2, and the worst was given a rank of 6 or 8, depending on the type of juice. The color assessment was the first stage of the evaluation and was carried out systematically under uniform lighting conditions to ensure the consistency and reliability of the results. Then, after the visual evaluation, the participants moved on to the second stage, which involved sensory evaluation of the samples based on odor. The third stage was a taste evaluation. During the tasting, participants could see the color of the juice.

#### 2.3.3. Color Measurements

Color measurements were obtained using a Tri-Color SF80 spectrophotometer, which was calibrated with standards L* 90.08, a* −0.74, b* 0.70. The measurement was conducted in SCI geometry, utilizing a D65 light source and an observation angle of 10°. A volume of 30 mL of the juice in question was transferred into a cuvette suitable for liquid analysis, and 15 measurements were taken, from which the arithmetic mean was subsequently calculated. Before each measurement of a new sample, the cuvette was carefully washed under running water and thoroughly dried.

In addition, after obtaining the CIELAB parameters, ∆E was calculated based on the following formula:$$\Delta E=\sqrt{{\left(L_{2}^{*}-L_{1}^{*}\right)}^{2}}+{\left(a_{2}^{*}-a_{1}^{*}\right)}^{2}+{\left(b_{2}^{*}-b_{1}^{*}\right)}^{2}$$

L*_1_ and L*_2_ are the lightness values for the first and second samples.a*_1_ and a*_2_ are the values of the a* (red–green) axis for both samples.b*_1_ and b*_2_ are the values of the b* (yellow–blue) axis for both samples.

### 2.4. Statistical Analysis

The data obtained from the color measurements were collated using the software programs Color QC (Narama, Poland, https://www.tri-color.pl/oprogramowanie/3color-color-qc-dla-spektrofotometrow, accessed on 3 December 2024) and Microsoft Office Excel 2021. The requisite statistical calculations were performed using the software package Statistica 13.3 (StatSoft, Krakow, Poland, 2017). The non-parametric ANOVA, Friedman’s test, and Wilcoxon’s paired order test were employed for the purpose of statistical analysis. The significance level was set at α = 0.05, while the Wilcoxon paired order test employed a Bonferroni correction to prevent a Type I error. A significance level of α = 0.0033 was adopted for beetroot juices and α = 0.0018 for tomato juices.

## 3. Results

### 3.1. Preference and Frequency of Consumption

The analysis of the frequency of juice consumption revealed that the most prevalent type of juice consumed was fruit juice. This was consumed on several occasions per month by 30% of respondents and on several occasions per week by 28%. The consumption of fruit and vegetable juices was markedly less frequent, with a significant proportion of respondents indicating that they did not consume such juices (32% and 44%, respectively). The majority of respondents (76%) did not consume beetroot juice, while 66% did not drink tomato juice. None of the respondents reported consuming juices other than fruit juices every day (Table 2). With regard to brand preferences, the most frequently selected brands of beetroot juices and tomato juices were Dawtona (16–20%) and Tarczyn (10%), respectively. It is noteworthy that brands such as Zaczarowany Ogród, NaturaVena, Bionaturo, Auchan, and Dolina Czerska were not selected by any respondent.

The results of the consumer preference survey indicate that 70% of respondents definitely like fruit juices, and 60% rather like fruit and vegetable juices. Vegetable juices were less popular, with 34% of respondents indicating that they rather dislike them (Table 3).

The primary factors that informed the selection of juice were taste (82%) and nutritional value (60%). Price was important for 42% of respondents, while the type and size of packaging were of the least importance to respondents. When questioned as to the rationale behind the consumption of vegetable juices, 32% of respondents cited nutritional value, while 18% indicated taste as the primary motivation. None of the respondents reported consuming vegetable juices as a result of a recommendation from a medical professional.

### 3.2. Instrumental Analysis of Beetroot Juice Color Using the Spectrophotometric Method in the CIELAB Color System

The color of the beetroot juice for the sample labeled DA was measured and found to have a value for the parameter L* ranging from 24.36 to 24.66. The parameter a* exhibited a range of 5.36 to 5.93, while the parameter b* demonstrated a range of 0.58 to 0.92. The mean values of the measurements for the juice labeled DA were as follows: the mean value of L* was 24.57 ± 0.09, while a* and b* exhibited mean values of 5.82 ± 0.18 and 0.81 ± 0.1, respectively.

With regard to the beetroot juice labeled TA, the values of the parameter L* were observed to range from 23.59 to 23.70. The parameter a* exhibited a range of 0.96 to 1.36, while the parameter b* demonstrated a range of −0.86 to −0.7.

The juice labeled RI exhibited a range of values for the L* parameter, spanning from 24.44 to 24.55. The a* parameter exhibited a range of values between 5.36 and 5.51, while the b* parameter demonstrated a range between 0.05 and 0.09.

With regard to the beetroot juice labeled ZO, the values of the L* parameter were observed to range from 24.05 to 24.1. The a* parameter exhibited a range of 3.18 to 3.4, while the b* parameter demonstrated a range of −0.27 to −0.18.

The results of the color measurements of NV beetroot juice in terms of the L* parameter exhibited values between 23.13 and 23.3. The a* measurement exhibited a range of values between 0.53 and 0.62, while the b* measurement demonstrated a range between −0.9 and −0.78.

With regard to the juice labeled SV, the values of the L* parameter were observed to range between 23.01 and 23.1. The a* parameter exhibited a range of values between 0.1 and 0.75, while the b* parameter demonstrated fluctuations between −0.88 and −0.8.

In comparison to the other samples, the DA juice exhibited the highest values for the L* parameter, whereas the SV juice displayed the lowest L* values. The highest values for the a* parameter were observed in the DA juice, while the lowest values were recorded for the NV juice. The b* parameter demonstrated a range of values from −0.83 to 0.81, with the highest values observed in the DA juice and the lowest values in the SV juice (Table 4).

Additionally, the mean values of the juice color measurements are illustrated in the chromaticity plots (Figure 3).

The analysis of the color of the juices revealed significant differences between the samples. The DA juice exhibited the highest values for the L* parameter, indicative of a lighter shade. The a* parameter also reached its highest value for DA juice, indicating that warm tones, primarily red, are dominant in its color. The lowest L* values were observed for SV juice, indicating its darker hue. The a* and b* parameters suggest the presence of more neutral red and blue tones.

The b* parameter, which ranged from −0.83 to 0.81, indicated that DA juice exhibited the highest level of yellow hue, while SV juice demonstrated the lowest value of this parameter, suggesting a predominance of blue hue in its color.

In conclusion, DA juice displayed the lightest color with a predominance of warm tones, while SV juice exhibited the darkest hue and a predominance of cool tones.

### 3.3. Instrumental Analysis of Tomato Juice Color Using the Spectrophotometric Method in the CIELAB Color System

The results of the color analysis of the tomato juices demonstrated the following values for the specified color parameters. The juice designated as DT exhibited L* parameter values that ranged from 30.72 to 30.83, a* values that ranged from 10.83 to 10.92, and b* values that ranged from 7.97 to 8.07. In comparison, the juice TC exhibited the highest L* parameter values, ranging from 33.61 to 33.77, and a* values of 10.12 to 10.19 and b* values of 8.81 to 8.95. The RV juice exhibited L* parameter values that ranged from 30.69 to 30.76, a* values that ranged from 9.48 to 9.54, and b* values that ranged from 6.75 to 6.84. The HT juice sample exhibited L* values of 31.26 to 31.49, a* parameters of 9.16 to 9.33, and b* values of 6.13 to 6.38. The TB juice exhibited L* values between 30.82 and 30.84, a* parameters ranging from 9.04 to 9.17, and b* values between 7.41 and 7.51. AU juice exhibited L* values ranging from 31.36 to 31.6, accompanied by a* parameters from 9.39 to 9.63 and b* values from 7.54 to 7.63. The BN juice sample exhibited L* values ranging from 30.48 to 30.64, a* values from 10.03 to 10.13, and b* values from 7.4 to 7.48. The L* values for DC juice ranged from 30.69 to 30.94, while the a* parameters ranged from 10.57 to 10.67 and the b* values ranged from 7.84 to 8.0.

A comparison of the mean values of the color measurements revealed that TC juice exhibited the highest and BN juice the lowest value of the L* parameter. The highest value of the a* parameter was observed in DT juice, while the lowest value of a* was found in TB juice. The b* parameter ranged from 6.22 to 8.9, with HT juice exhibiting the lowest b* value and TC juice achieving the highest b* value (Table 5).

Additionally, the mean values of the juice color measurements are illustrated in the chromaticity plots (Figure 4).

A summary of the results of the color measurements of the tomato juices indicates significant differences in the color parameters of the samples analyzed. The TC tomato juice exhibited the highest values for the L* parameter, indicating the greatest brightness among all samples. In contrast, the BN juice displayed the lowest L* values, suggesting a darker hue.

With regard to the a* parameter, which determines the intensity of red, the DT juice exhibited the highest values, indicating a stronger red hue, whereas the TB juice displayed the lowest values, suggesting a less intense red color. The values of the b* parameter, corresponding to the yellow tone, ranged from 6.22 to 8.9. The juice sample designated HT exhibited the lowest b* value, indicating a dominance of cooler, bluer tones. In contrast, the TC juice sample reached the highest b* value, indicating a more pronounced presence of yellow tones in its color.

These findings suggest that the TC juice sample was the most light and dominated by yellow tones, while the HT juice sample had a darker color with a predominance of blue tones.

### 3.4. Sensory Analysis of the Color, Odor, and Taste of Beetroot Juice by the Serialization Method

A statistical analysis of the organoleptic evaluation of color revealed significant differences in the acceptability ratings provided by the study participants (*p* < 0.0001). The SV brand of beetroot juice was rated the highest in terms of color, while the NV brand was rated second. In contrast, the DA brand of juice was rated the lowest in terms of color. The Kendall’s concordance coefficient (Wk = 0.443) indicates moderate agreement between the respondents’ ratings of the color of the beetroot juice samples. Furthermore, a comparison of the color acceptability ratings with the spectrophotometric measurements indicated that the beet juice with the darkest color and the bluest hue was the most acceptable. In contrast, the juice with the lightest color and the most yellow hue received the lowest ratings from respondents (Table 6).

Table 7 presents the results of statistical analysis conducted using the Wilcoxon signed-rank test, showing differences between the analyzed data pairs for beetroot juice and parameters related to color.

A statistical analysis of the organoleptic ratings for odor revealed significant differences in levels of acceptability among respondents (*p* = 0.00001). The beetroot juice of the TA brand was rated the highest for odor (rank sum 118), while the NV juice was rated the second highest (rank sum 164). The DA brand of juice was assigned the lowest score for odor (rank sum 213). The value of Kendall’s concordance coefficient (Wk = 0.122) indicates a low level of agreement among respondents regarding the perceived odor of the beetroot juice samples. A comparative analysis of the odor acceptability results with spectrophotometric measurements revealed that the juice with the lightest color was the least acceptable in terms of odor. However, no correlation was observed between color brightness and respondents’ odor preference for the other juices (Table 8).

The statistical differences observed between paired data for the odor parameters of beetroot juice have been detailed in Table 9, based on the results of the Wilcoxon signed-rank test.

Statistical analysis of the organoleptic ratings for taste showed significant differences in the level of acceptance of the beetroot juices among the study participants (*p* < 0.00001). Juice of the TA brand was rated highest for taste, while juice of the RI brand was ranked second, with a rank sum of 135. Juices of the NV and ZO brands received the lowest ratings for taste. The value of Kendall’s concordance coefficient (Wk = 0.176) indicates a low level of concordance of respondents’ answers in the taste evaluation of the samples analyzed. Furthermore, a comparative analysis of the taste acceptability results with the data obtained from the spectrophotometric measurements showed no correlation between the taste qualities of the juices and their color parameters (Table 10).

Table 11 presents the outcomes of the Wilcoxon signed-rank test, outlining the statistical differences between paired data for beetroot juice taste parameters.

### 3.5. Sensory Analysis of the Color, Odor, and Taste of Tomato Juice by the Serialization Method

A statistical analysis of the organoleptic ratings for the color of the tomato juices revealed significant differences in acceptance between respondents (*p* < 0.0001). The BN brand juice was rated the highest for color, ranking first, while the DT and RV brand juices were rated second and third, respectively. The TC brand juice was assigned the lowest rating for color, with a rank sum of 353. The Kendall’s concordance coefficient value (Wk = 0.356) indicates a low level of concordance in color ratings among the study participants. A comparison of the color acceptability results with spectrophotometric measurements revealed that the tomato juice with the highest acceptability was characterized by the darkest hue, while the juice with the lowest acceptability exhibited the lightest color with a dominant yellow hue (Table 12).

The statistical differences between paired data for the taste parameters of tomato juice are illustrated in Table 13, based on the results of the Wilcoxon signed-rank test.

A statistical analysis of the organoleptic evaluation of the aroma of the tomato juices revealed significant differences in acceptance among respondents (*p* = 0.00008). The DC brand tomato juice was rated the highest for aroma, ranking first, while the RV and AU juices were rated second and third, respectively. The TC brand juice was assigned the lowest score for odor, with a rank sum of 278. The Kendall’s concordance coefficient value (Wk = 0.087) indicates a low level of concordance in the odor ratings among the study participants. A comparison of the odor ratings with the spectrophotometric measurements revealed that the juice with the least acceptable odor had the brightest color. However, for the other juices, no correlation was found between color brightness, as measured by spectrophotometry, and respondents’ odor preferences (Table 14).

The results of the Wilcoxon signed-rank test, highlighting the statistical differences between paired data for the odor parameters of tomato juice, are presented in Table 15.

A statistical analysis of the organoleptic taste ratings of the tomato juices revealed significant differences in acceptance among the study participants (*p* < 0.00001). The AU brand of juice was rated the highest for flavor, with the HT brand ranking second and the DC brand ranking third. The BN brand juice was assigned the lowest taste scores. The value of Kendall’s concordance coefficient (Wk = 0.116) indicates a low level of consistency in the survey participants’ responses regarding the taste ratings of the samples. A comparative analysis of the spectrophotometric results with the taste ratings of the juices revealed no significant correlation between color parameters and taste qualities (Table 16).

Table 17 presents the results of the Wilcoxon signed-rank test, detailing the statistical differences between paired data for the taste parameters of tomato juice.

## 4. Discussion

The results of the survey indicated a relatively low consumption of vegetable juices, whereas fruit juices were selected with greater frequency by consumers. Similar results were obtained by Pyryt and Karpińska, also indicating a low consumption of vegetable juices [42]. In our own study, the most preferred brands of tomato and beetroot juices were Dawton and Tarczyn, while in the study by Pyryt and Karpińska [42], the leaders of vegetable juice brands were Tymbark and Hortex. These differences may result from the form of packaging, because Dawton and Tarczyn juices are sold in small glass bottles, which is practical in the case of more specific flavors. Among the factors influencing the choice of juices, taste (82%), nutritional value (60%), and price (42%) were the most important for respondents, while the type and size of packaging (10%) were of the least importance. Similar results were obtained in the study by Pyryt and Karpińska [42], except that price (29%) was more important than taste (26%), and the quality of the product (40%) was the most important. In the cited study, the packaging was also of marginal importance to consumers (5%). In turn, the study by Balon et al. [43] showed that the size of the packaging was important for 75% of respondents, but the quality of the product and taste were still key criteria for over 90% of respondents. When asked about the reasons for consuming vegetable juices, 32% of respondents indicated the nutritional value, while 18% indicated taste. However, different results were obtained by Pyryt and Karpińska [42], in which the respondents indicated taste first (62%), followed by health benefits associated with drinking juice. These differences may result from the characteristics of the study group, because the participants in our study were mainly dieticians.

The sensory tests of juices demonstrated notable discrepancies in the evaluations of specific attributes, which could be attributed to variations in storage, processing, and product composition. The analysis of the results demonstrated that there was no correlation between the taste assessment and preferences for color and odor. For instance, the BN brand of tomato juice received the highest score in the color category but the lowest score in the taste assessment. Similarly, the DA brand of beetroot juice, despite receiving the lowest scores for color and odor, was classified in third place in terms of taste. It is also noteworthy that the brands identified as the most frequently selected by consumers in our own survey did not perform the best in the organoleptic assessment.

The study conducted by Zhu et al. [44] was designed to compare consumer preferences for freshly squeezed tomato juice with those for commercial juice. The findings revealed notable discrepancies between the two products, with consumers demonstrating a proclivity to pay a premium for the juice that most closely approximated the taste of fresh tomatoes. This suggests that taste plays a pivotal role in influencing consumer preferences. Further observations indicated that, although the commercial juice was rated more highly in terms of appearance prior to tasting, after tasting, consumers expressed a clear preference for the freshly squeezed juice. Similar outcomes were observed in the study conducted by Koltun et al. [45], wherein consumers initially rated the commercial juice higher, but subsequent to the taste test, the juice derived from fresh tomatoes received higher scores. Our own studies also demonstrated a lack of consistency between the assessment of juice color and its taste qualities, indicating that consumer perception of the product is determined by different sensory aspects. In light of these findings, it is recommended that manufacturers give special attention to both the visual appeal and taste of their products. Enhancing both the color and taste quality of juices can significantly increase their attractiveness to consumers and contribute to achieving a more favorable market position.

In the study conducted by Iijima et al. [46], the influence of volatile compounds on the perception of taste of processed juices available on the Japanese market was analyzed in comparison to fresh tomato juices. The findings indicated that processed juices were perceived as more palatable than fresh juices, which also suggested that the presence of certain volatile compounds may significantly influence taste perception. In our own study, only TA beetroot juice obtained the highest scores for both odor and taste, while in the case of the other juices, no significant correlations between odor and taste were observed.

The selection of food products by consumers is frequently influenced by their visual appeal, as the evaluation of sensory attributes such as taste or aroma can only occur subsequent to the purchase. Consequently, an analysis of product color preferences is of particular importance for manufacturers seeking to attract potential buyers. The study conducted by Savaş Bahçeci et al. [47] demonstrated that the optimal color for tomato products should be characterized by a high level of red and a minimum amount of yellow tones. The findings of our own research indicated that tomato juice with the highest yellow tone content received the lowest ratings in terms of color. Conversely, the juice with the highest red intensity, despite being rated more favorably than the juice with a predominance of yellow tones, was nonetheless placed second in the ranking of ratings. In the analysis of beetroot juices, it was also observed that the juice with the most yellow shade was rated the worst in terms of color. These results are consistent with the findings of the research conducted by Savaş Bahçeci et al. [47], which suggest that products such as tomato juices should present a high red intensity and a low content of yellow tones to optimize consumer acceptance.

The distinctive color properties and high content of bioactive compounds present in beetroot make it an essential ingredient in the production of modern functional beverages. The research conducted by Kayın et al. [48] demonstrated that the color stability of beetroot juice is closely associated with storage time and temperature. This was evaluated using the CIELAB system. Furthermore, the studies conducted by Tobolkov et al. [36] highlighted that the primary factors influencing the color of beetroot juice are the pigments present, particularly betalains, in addition to the processing techniques employed, including heat treatment. Similar conclusions were reached by Yin Zhang et al. [49], who examined the correlation between the color, aroma, and flavor of juice and the results of color measurements in the CIELAB system. In their studies, it was observed that the stability of the juice’s sensory attributes exhibited a decline after 30 min of storage at room temperature. Similar outcomes were observed in the studies conducted by Bianchi et al. [23] and Tobolkov et al. [36], who also corroborated the impact of heat treatment and storage on the alteration in juice color. The findings of the aforementioned studies indicate that instrumental color measurements are an effective method for monitoring and controlling the quality of food products, including juices. While our own studies did not consider the impact of storage time on color, a notable correlation was observed between the results of color measurements and the subjective sensory perceptions of consumers. This emphasizes the necessity for manufacturers to integrate these factors in order to enhance the quality and acceptance of their products by consumers.

The analysis of the color of juices in the CIELAB color space, as conducted in various studies, provides valuable information on the influence of factors such as production method, type of raw material, and storage time on the final color of the products. In a study conducted by Tobolkova et al. [36], the CIELAB parameter values for beetroot juices were analyzed. The results demonstrated that: the L* value ranged from 16.12 to 16.75, while the a* value ranged from 3.76 to 4.78, and the b* value ranged from 0.60 to 1.09. In our own studies, the values observed for beetroot juices were found to be higher. The L* value ranged from 23.07 to 24.57, the a* value from 0.57 to 5.82, and the b* value from −0.83 to 0.81. Additionally, Zhang et al. [49] conducted studies on the evaluation of product color using the CIELAB color space. The researchers analyzed freshly squeezed juices obtained from six commonly consumed types of fruit and vegetables. One of the juices subjected to analysis was tomato juice. The products were subjected to repeated analyses at designated time points, with the final assessment conducted 420 min after the initial juicing process. The color values of tomato juice were measured immediately after juicing and found to be 65.27 for L*, −28.93 for a*, and 58.58 for b*. These values underwent a change over time, with the juice exhibiting the following color values at 420 min after extraction: L* = 77.35, a* = −9.83, and b* = 18.92. A comparison of these data with the study by Jeż et al. [50], which measured the color of tomato juices made from fresh fruits, revealed that the L* values ranged from 29.39 to 34.80, the a* values from 17.20 to 24.04, and the b* values from 24.34 to 30.31. Our study did not include freshly squeezed juices, but the measurement results for tomato juices ranged from 23.07 to 24.57 for L*, from 0.57 to 5.82 for a*, and from −0.83 to 0.81 for b*. Additionally, the analysis of tomato juice color was conducted by Łupina and Kowalczyk [51]. The study encompassed nine juices derived from conventional cultivation and two from organic cultivation. The measured juice color values ranged from 35.95 to 38.63 for L*, from 7.70 to 13.43 for a*, and from 5.86 to 7.84 for b*. In comparison with our results, it can be concluded that the juice production method, cultivation method, and storage time significantly affect the color parameters. The findings of these studies collectively suggest that monitoring and controlling color during production may play a pivotal role in ensuring the desired sensory characteristics of juices.

The study conducted by Bielaszka et al. [52] on the analysis of five different tomato juices included both color measurements and a sensory evaluation. The sensory analysis of color conducted among dietitians revealed that the juice with the most intense red color was perceived as the most acceptable in terms of color. In the context of our own study, the darkest tomato juice was rated the highest for color. The results of both the Bielaszka et al. [52] study and our own analyses indicated that the juice with the lightest and most yellow shade was the least preferred in terms of color. However, the study by Łupina and Kowalczyk [51] yielded disparate results, with the lightest juice receiving the highest evaluation of color from a ten-person trained sensory panel. These discrepancies may be attributed to varying aesthetic and sensory preferences among different consumer groups, underscoring the necessity of adapting products to align with specific market expectations. It is crucial for food manufacturers to consider the color preferences of their target groups when designing products. For the dietary market segment, where the primary selection criterion is nutritional value, manufacturers must ensure that color modifications do not negatively impact the nutritional aspects of the products.

The measurement of instrumental color can be used to ascertain the presence of specific chemical compounds in food products, which have a significant impact on their quality and visual appeal. In the context of tomato juices, lycopene plays a significant role in determining the intensity of the red hue. The studies conducted by Łupina and Kowalczyk [51] demonstrated a significant correlation between lycopene concentration and the intensity of red in tomato juices, thereby underscoring its pivotal role in defining their color. Similarly, betalains are the primary compounds responsible for the distinctive red–violet hue observed in beetroot juices. The studies conducted by Virginia Prieto-Santiago and colleagues demonstrated that the relationship between color and total betalain content varies depending on the type of processed product, with the closest correlations occurring in juices. The studies conducted by Tobolkova et al. [36] and Bianchi et al. [23] highlight the pivotal role of betalains in conferring the intense color of beetroot juice. It is evident that processes such as storage and pasteurization can influence the content of these pigments and consequently modify the color of juices. All of the studies discussed employed the CIELAB color system for the assessment of colors, which is consistent with the methodology used in our own studies. The findings of these analyses indicate that processes such as storage and pasteurization may result in a reduction in betalain content and a modification of the juice’s color. Consequently, extending our own studies to determine the content of betalains in beetroot juices and lycopene in tomato juices could provide additional information on the quality and health-promoting values of these products. Such studies could contribute to a better understanding of the impact of technological processes on the sensory and health properties of juices, which is important for both producers and consumers looking for healthy food products.

It is important to consider the limitations of the sensory analysis, which may influence the interpretation of the results and inform the design of future studies. A significant limitation of the study was the narrow age range of the participants, with the majority falling within the 23–24 age bracket. It would be beneficial for future studies to expand the age range of participants to include individuals from different age groups. This would allow for a comparison of the results with the analyses conducted in this work and for the collection of more comprehensive data. Furthermore, the gender composition of the participant group, which predominantly consisted of women, reflects the demographic structure of dietitians in Poland, where women form the vast majority in this profession. While this accurately represents the target population, it also limits the generalizability of the findings to a more diverse demographic. Future studies should aim to include a more balanced participant pool to provide broader insights into consumer preferences. Furthermore, the sensory analysis did not take into account the price of the products or the detailed information included on the label of the analyzed product. These aspects are of particular importance to the group of dietitians and consumers who, when choosing food products, are guided not only by their sensory qualities but also by their nutritional profile. The integration of these variables in future studies could enrich the analysis and provide a more comprehensive picture of the evaluation of food products.

The findings of this study demonstrate clear differences in sensory evaluations of beetroot and tomato juices, with darker-colored juices generally receiving higher ratings for color. However, as highlighted in the review, the connection between instrumental color parameters and sensory attributes such as taste and aroma is not always straightforward. For instance, juices with high acceptability in color did not necessarily receive equivalent ratings in taste or aroma. These discrepancies suggest that consumer preferences are influenced by a combination of sensory and contextual factors.

To address this complexity, future studies should incorporate additional variables, including cultural and aesthetic influences, which may shape consumer perceptions of sensory attributes. For instance, differences in regional or cultural preferences for certain color profiles could play a role in determining the acceptance of specific juice products. Similarly, contextual factors such as product presentation or packaging might interact with sensory attributes to influence consumer preferences.

Moreover, the inclusion of attributes like texture, which were not evaluated in this study, could provide a more comprehensive understanding of consumer acceptance. Texture, in combination with color, aroma, and flavor, forms a holistic sensory experience that might better predict consumer preferences. Extending the demographic diversity of participants beyond dietitians would also provide a broader perspective, allowing for a more generalizable set of recommendations for manufacturers.

A notable benefit of the sensory studies conducted was the laboratory’s adherence to the PN-EN ISO 8589:2010 standard [36]. Adherence to these standards guaranteed the precision of the assessments and facilitated the acquisition of reproducible outcomes, which is pivotal for the credibility of the studies. The color analysis was conducted using both objective and subjective methods. The objective methods, including color measurements using instrumental systems, yielded precise data on color parameters, whereas the subjective methods, such as those conducted by a sensory panel, permitted the assessment of consumers’ color perception. This combination of methodologies provides a comprehensive account of the analyzed products, taking into account both precise instrumental data and the subjective perceptions of the recipients. From a marketing perspective, it was also of importance to ascertain the precise color preferences of juices, as these can have a significant impact on marketing strategies and product design. A precise understanding of consumers’ color preferences can enable manufacturers to adapt their products to market expectations more effectively, which can contribute to their better acceptance by recipients. The appropriate selection of the group of respondents whose preferences were studied is also important, because it can influence dietary decisions in the wider society. The results of such studies can contribute to more effective product development and market strategy optimization.

## 5. Conclusions

The results of the survey indicated that consumers exhibit a greater preference for and consume fruit juices more frequently than vegetable juices or fruit and vegetable juices. Based on the conducted sensory studies, significant differences were found between the analyzed tomato juices and beetroot juices in terms of color, odor, and taste. The analysis did not show that any of the tested juices obtained the highest scores in all of these categories. The spectrophotometric measurement of color, conducted using the CIELAB system, demonstrated that juices with a darker hue received higher ratings in sensory evaluations than those with a lighter color.

The dietitians’ preferences, which indicated higher scores for darker juices, provide a valuable clue for food manufacturers. These findings suggest that adjusting the composition of juices to enhance darker hues, potentially by optimizing the concentration of color-determining compounds such as betalains and lycopene, may improve consumer acceptance. Additionally, producers are encouraged to focus on balancing visual appeal with sensory attributes like taste and aroma to align with consumer preferences.

Furthermore, the findings underline the importance of integrating sensory analysis with instrumental color measurements in the design of new products. Future product development could incorporate these insights to create juices that harmonize visual and sensory qualities, offering both aesthetic appeal and flavor satisfaction. Consideration of contextual factors such as packaging design and marketing strategies that emphasize these sensory attributes could further enhance consumer engagement and product marketability.

Finally, the study highlights the need to expand future research to include more diverse demographic groups and access to product labels, ensuring the generalizability of the findings and a deeper understanding of consumer behavior. These efforts will support manufacturers in developing health-promoting beverages tailored to a broader spectrum of consumer preferences.

## Figures and Tables

**Figure 1 foods-13-04059-f001:**
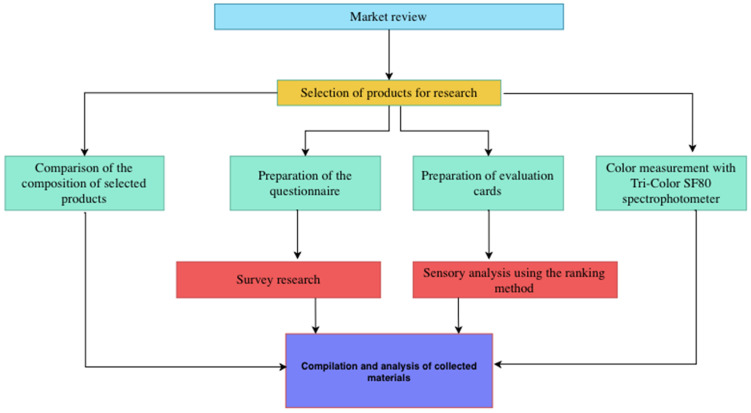
Scheme of conduct of the study.

**Figure 2 foods-13-04059-f002:**
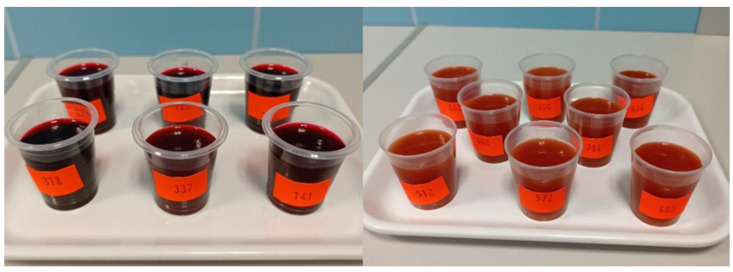
Coded samples of beetroot and tomato juices.

**Figure 3 foods-13-04059-f003:**
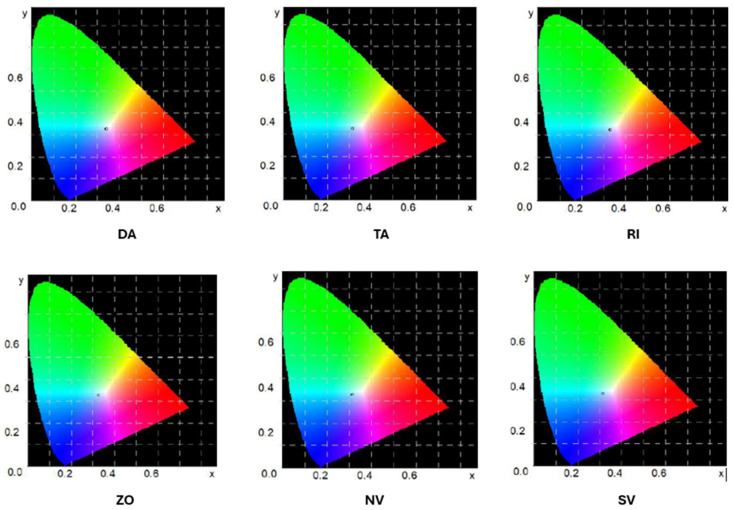
Beetroot juice color measurement values on chromaticity charts (Green region: Higher representation of green wavelengths; Red region: Dominance of red wavelengths; Blue region: Higher intensity of blue wavelengths; White center: Neutral or balanced light).

**Figure 4 foods-13-04059-f004:**
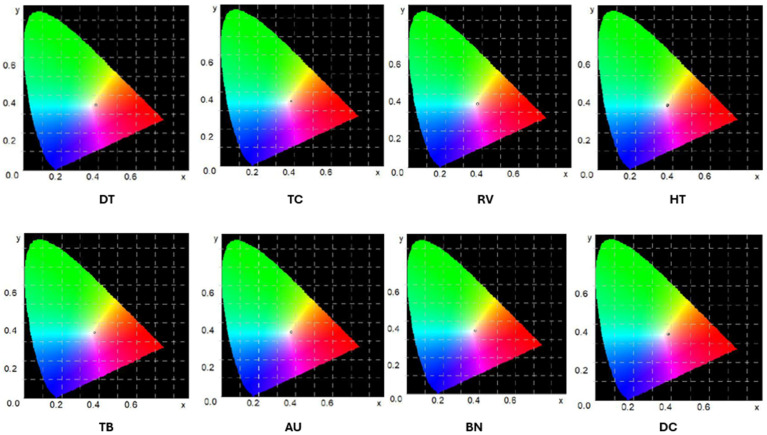
Tomato juice color measurement values on chromaticity charts (Green region: Higher representation of green wavelengths; Red region: Dominance of red wavelengths; Blue region: Higher intensity of blue wavelengths; White center: Neutral or balanced light).

**Table 1 foods-13-04059-t001:** Characteristics of the tested juices.

Name Juice	Sample Code	Composition and Method of Production	Energy Value in 100 mL [kcal]	Salt in 100 mL [g]	City, Country
Beetroot juices					
DA	337	Juices reconstituted from concentrated juices and purees of: beetroot (87%), apples (10%), celery, lemons, ginger, garlic, salt, spices	32	0.48	Błonie, Poland
TA	214	Juices from concentrated juices of: beetroot (87%), apples (12%), lemons, sea salt, spices and spice extracts	37	0.39	Tychy, Poland
RI	141	Beetroot juice from beetroot juice concentrate (82%), apple juice from apple juice concentrate (10%), beetroot puree (5%), water, salt (0.5%), acidity regulator—citric acid, spices	44	0.6	Myszków, Poland
ZO	273	Beetroot juice from beetroot juice concentrate (82%), apple juice from apple juice concentrate (10%), beetroot puree (5%), water, salt (0.5%), acidity regulator—citric acid, spices	44	0.7	Dąbrowa Chełmińska, Poland
NV	318	Juice reconstituted from concentrated beetroot juice, antioxidant: ascorbic acid, acidity regulator—citric acid	45	0.1	Leszno, Poland
SV	123	Beetroot juice from concentrated juice (99.3%), salt	64	0.79	Wałcz, Poland
Tomato juices					
DT	485	Tomato juice reconstituted from concentrated juice 99.5%, salt	19	0.4	Błonie, Poland
TC	654	Tomato juice, sea salt	19	0.65	Tychy, Poland
RV	660	Tomato juice from concentrate, sea salt	17	0.5	Myszków, Poland
HT	572	Tomato juice (100%) from concentrated juice, salt	17	0.6	Warsaw, Poland
TB	512	Tomato juice from concentrate, tomato juice, sea salt, chili—cayenne pepper, black pepper	17	0.64	Tychy, Poland
AU	784	100% tomato juice from raspberry tomatoes	19	1	Góra Kalwaria, Poland
BN	404	Organic tomato juice (99.74%)—reconstituted from concentrate, sea salt	18	0.26	Lublin, Poland
DC	863	99% tomato juice, salt	19	1	Góra Kalwaria, Poland

**Table 2 foods-13-04059-t002:** Frequency of consumption of different types of juices by respondents N = 50 (100%).

Type of Juice	Respondents’ Responses				
	I Don’t Consume	Once a Month	A Few Times a Month	A Few Times a Week	Every Day
	N (%)	N (%)	N (%)	N (%)	N (%)
Fruit juices	6 (12)	13 (26)	15 (30)	14 (28)	2 (4)
Fruit and vegetable juices	16 (32)	15 (30)	10 (20)	9 (18)	0 (0)
Vegetable juices	22 (44)	16 (32)	8 (16)	4 (8)	0 (0)
Beetroot juice	38 (76)	7 (14)	3 (6)	2 (4)	0 (0)
Tomato juice	33 (66)	7 (14)	9 (18)	1 (2)	0 (0)

**Table 3 foods-13-04059-t003:** Preferences for consumption of different types of juices by respondents N = 50 (100%).

Type of Juice	Respondents’ Responses				
	I Definitely Don’t Like It	I Rather Don’t Like It	I Have no Opinion	I Rather Like It	I Definitely Like It
	N (%)	N (%)	N (%)	N (%)	N (%)
Fruit juices	2 (4)	0 (0)	0 (0)	13 (26)	35 (70)
Fruit and vegetable juices	3 (6)	6 (12)	3 (6)	30 (60)	8 (16)
Vegetable juices	6 (12)	17 (34)	9 (18)	15 (30)	3 (6)

**Table 4 foods-13-04059-t004:** Comparison of the mean values of beetroot juice measurements.

Juice Name	L* (X ± SD)	a* (X ± SD)	b* (X ± SD)
DA	24.57 ± 0.09	5.82 ± 0.18	0.81 ± 0.1
TA	23.66 ± 0.04	1.14 ± 0.16	−0.77 ± 0.05
RI	24.49 ± 0.04	5.43± 0.04	0.07 ± 0.01
ZO	24.08 ± 0.02	3.32 ± 0.08	−0.22 ± 0.03
NV	23.2 ± 0.06	0.57 ± 0.03	−0.82 ± 0.04
SV	23.07 ± 0.02	0.74 ± 0.01	−0.83 ± 0.02
*p*-value	<0.001 *	<0.001 *	<0.001 *

X = average; SD = standard deviation; * = *p* < 0.05; L* = lightness; a* = green–red tones; b* = blue–yellow tones.

**Table 5 foods-13-04059-t005:** Comparison of the mean values of tomato juice measurements.

Juice Name	L* (X ± SD)	a* (X ± SD)	b* (X ± SD)
DT	30.76 ± 0.03	10.89 ± 0.03	8.04 ± 0.02
TC	33.67 ± 0.06	10.15 ± 0.02	8.9 ± 0.04
RV	30.71± 0.02	9.52 ± 0.02	6.8 ± 0.03
HT	31.39 ± 0.07	9.22 ± 0.05	6.22 ± 0.07
TB	30.83 ± 0.01	9.08 ± 0.03	7.44 ± 0.02
AU	31.56 ± 0.07	9.42 ± 0.04	7.59 ± 0.03
BN	30.58 ± 0.04	10.07 ± 0.03	7.43 ± 0.02
DC	30.89 ± 0.08	10.6 ± 0.03	7.88 ± 0.05
*p*-value	<0.001 *	<0.001 *	<0.001 *

X = average; SD = standard deviation; * = *p* < 0.05; L* = lightness; a* = green–red tones; b* = blue–yellow tones.

**Table 6 foods-13-04059-t006:** Summary of the results of color evaluation of beetroot juices by the serialization method.

Juice Name	X ± SD	Med	Sum of Ranks	Juice Quality Order	*p*-Value
DA	5.6 ± 1.07	6	280	VI	<0.0001 *
TA	2.82 ± 1.17	3	141	III	
RI	3.68 ± 1.45	4	184	IV	
ZO	4.1 ± 1.2	4.5	205	V	
NV	2.58 ± 1.47	2	129	II	
SV	2.22 ± 1.31	2	111	I	

X = average; SD = standard deviation; Med = median; * = *p* < 0.05.

**Table 7 foods-13-04059-t007:** Results of Wilcoxon Signed-Rank Test for Beetroot Juice Color Parameters.

	DA	TA	RI	ZO	NV	SV
DA	x	0.002398				
TA	0.002398	x				
RI	0.004515	0.308479	x			
ZO	0.024808	0.000010	0.000021	x		
NV	0.110129	0.000025	0.000273	0.980746	x	
SV	0.505363	0.000317	0.003554	0.124816	0.160155	x
LEGEND:	*p* > 0.05	*p* > 0.05	*p* < 0.005	*p* < 0.001		

**Table 8 foods-13-04059-t008:** Summary of the results of odor evaluation of beetroot juice.

Juice Name	X ± SD	Med	Sum of Ranks	Juice Quality Order
DA	3.84 ± 1.28	5	213	VI
TA	2.36 ± 1.69	1.5	118	I
RI	3.48 ± 1.64	3.5	174	III
ZO	3.77 ± 1.66	4	189	IV
NV	3.28 ± 1.67	3	164	II
SV	3.84 ± 1.28	4	192	V

X = average; SD = standard deviation; Med = median.

**Table 9 foods-13-04059-t009:** Results of Wilcoxon Signed-Rank Test for Beetroot Juice Odor Parameters.

	DA	TA	RI	ZO	NV	SV
DA	x	0.000059				
TA	0.000059	x				
RI	0.023298	0.003340	x	0.015395		
ZO	0.174526	0.001181	0.281776	x		
NV	0.014792	0.015395	0.717354	0.142296	x	
SV	0.248679	0.000104	0.329570	0.881060	0.072573	x
LEGEND:	*p* > 0.05	*p* > 0.01	*p* < 0.005	*p* < 0.001		

**Table 10 foods-13-04059-t010:** Summary of beetroot juice taste evaluation results.

Juice Name	X ± SD	Med	Sum of Ranks	Juice Quality Order
DA	3.56 ± 1.68	3	178	III
TA	2.42 ± 1.1.46	2	121	I
RI	2.7 ± 1.36	3	135	II
ZO	4.32 ± 1.48	5	216	VI
NV	4.22 ± 1.79	5	211	V
SV	3.78 ± 1.59	4	189	IV

X = average; SD = standard deviation; Med = median.

**Table 11 foods-13-04059-t011:** Results of Wilcoxon Signed-Rank Test for Beetroot Juice Taste Parameters.

	DA	TA	RI	ZO	NV	SV
DA	x	0.002398				
TA	0.002398	x				
RI	0.005515	0.308479	x			
ZO	0.024808	0.000010	0.000021	x		
NV	0.110129	0.000025	0.000273	0.980746	x	
SV	0.505363	0.000317	0.003554	0.124816	0.160155	x
LEGEND:	*p* > 0.05	*p* > 0.01	*p* < 0.005	*p* < 0.001		

**Table 12 foods-13-04059-t012:** Summary of the results of color evaluation of tomato juices by the serialization method.

Juice Name	X ± SD	Med	Sum of Ranks	Juice Quality Order
DT	3.2 ± 1.68	3	160	II
TC	7.06 ± 1.85	8	353	VIII
RV	3.7 ± 1.94	4	185	III
HT	4.54 ± 1.78	5	227	V
TB	5.36 ± 1.95	6	268	VI
AU	5.7 ± 1.83	6	285	VII
BN	2.66 ± 2.17	2	133	I
DC	3.78 ± 1.61	4	189	IV

X = average; SD = standard deviation; Med = median.

**Table 13 foods-13-04059-t013:** Results of Wilcoxon Signed-Rank Test for Tomato Juice Color Parameters.

	DT	TC	RV	HT	TB	AU
DT	x	0.000000				
TC	0.000000	x				
RV	0.204297	0.000001	x			
HT	0.001445	0.000001	0.032112	x		
TB	0.000014	0.000027	0.001285	0.052932	x	
AU	0.000009	0.000160	0.000094	0.007497	0.349084	x
BN	0.078117	0.000000	0.021319	0.000244	0.000015	0.000003
LEGEND:	*p* > 0.05	*p* > 0.01	*p* < 0.005	*p* < 0.001		

**Table 14 foods-13-04059-t014:** Summary of the results of odor evaluation of tomato juice.

Juice Name	X ± SD	Med	Sum of Ranks	Juice Quality Order
DT	5.18 ± 2.24	6	259	VII
TC	5.56 ± 2.37	6	278	VIII
RV	3.82 ± 1.92	4	191	II
HT	4.48 ± 2.32	4	224	V
TB	4.40 ± 1.91	4	220	IV
AU	4.28 ± 2.17	4	214	III
BN	4.94 ± 2.55	5	247	VI
DC	3.34 ± 2.12	3	167	I

X = average; SD = standard deviation; Med = median.

**Table 15 foods-13-04059-t015:** Results of Wilcoxon Signed-Rank Test for Tomato Juice Odor Parameters.

	DT	TC	RV	HT	TB	AU	BN
DT	x						
TC	0.336804	x					
RV	0.004272	0.001161	x				
HT	0.137120	0.055961	0.118996	x			
TB	0.105896	0.016019	0.195826	0.839355	x		
AU	0.063134	0.015395	0.324804	0.685156	0.854475	x	
BN	0.643109	0.299398	0.015191	0.428613	0.294925	0.252660	x
DC	0.001596	0.000056	0.195826	0.021869	0.016232	0.018264	0.003958
LEGEND:	*p* > 0.05	*p* > 0.01	*p* < 0.005	*p* < 0.001			

**Table 16 foods-13-04059-t016:** Summary of tomato juice taste evaluation results.

Juice Name	X ± SD	Med	Sum of Ranks	Juice Quality Order
DT	4.56 ± 2.10	4.5	228	V
TC	5.20 ± 2.40	6	260	VII
RV	4.50 ± 2.00	5	225	IV
HT	3.64 ± 1.86	4	182	II
TB	4.88 ± 2.43	5.5	244	VI
AU	3.40 ± 1.93	3	170	I
BN	5.90 ± 2.62	7.5	295	VIII
DC	3.92 ± 1.94	3	196	III

X = average; SD = standard deviation; Med = median.

**Table 17 foods-13-04059-t017:** Results of Wilcoxon Signed-Rank Test for Tomato Juice Taste Parameters.

	DT	TC	RV	HT	TB	AU	BN
DT	x						
TC	0.220212	x					
RV	0.746404	0.079763	x				
HT	0.027401	0.003780	0.027740	x			
TB	0.520911	0.517781	0.508453	0.008406	x		
AU	0.014595	0.000655	0.024808	0.351574	0.001048	x	
BN	0.011752	0.138400	0.006391	0.000202	0.089324	0.000041	x
DC	0.126007	0.009151	0.216601	0.565718	0.061106	0.111208	0.000167
LEGEND:	*p* > 0.05	*p* > 0.01	*p* < 0.005	*p* < 0.001			

## Data Availability

The original contributions presented in this study are included in the article. Further inquiries can be directed to the corresponding author.

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
