# Peer review of "Consumer Preferences, Sensory Evaluation, and Color Analysis of Beetroot and Tomato Juices: Implications for Product Development and Marketing in Health-Promoting Beverages"

_foods, 2024, doi:10.3390/foods13244059_

Round 1

Reviewer 1 Report

Comments and Suggestions for Authors

Author Response

Thank you so much for taking the time to evaluate our work. We have tried to incorporate all your valuable suggestions. If we could improve our work in any way, please let us know.

Comment 1

The title is clear and specific. It precisely indicates the aspects being studied, such as consumer preferences, sensory evaluation and colour analysis, and in which context the study is being carried out, i.e. on beet and tomato juices.

Thank you for your comment.

Comment 2

The abstract of the article serves its purpose of clearly presenting the objectives, methodology, main results and conclusions of the study. However, it could be improved by including more specific details on how colour parameters influence consumer perception. This would allow the practical applicability of the findings to be highlighted more clearly and better aligned with the needs of functional food producers.

Corrected as suggested by the reviewer.

Comment 3

The introduction does a good job of contextualizing the relevance of functional foods, particularly beet and tomato juices, as sources of bioactive compounds. However, the literature review could be enriched with more recent studies linking instrumental color analysis to sensory perception. Furthermore, the introduction lacks a robust theoretical framework explaining how these sensory attributes impact consumer decisions, which is essential for product development in the healthy food market.

Corrected as suggested by the reviewer.

Comment 4

It would be highly advisable to include hypotheses in the article, especially since a study involving sensory analysis and instrumental measurement of parameters (such as color) can benefit from clear hypotheses that guide the approach and allow for a more structured interpretation of the results. Hypotheses provide a clear direction to the research and better connect it to the theories or background presented in the introduction, the results can be discussed more clearly based on whether they confirm or refute the hypotheses put forward, and they help readers quickly identify the purpose of the analyses and how they relate to the objectives of the study..

Added description as suggested.

Comment 5

The materials and methods are well detailed, including the description of the study design, measurement tools, and statistical analysis. The inclusion of normative methods such as the CIELAB system and the laboratory under ISO standards reinforces the technical validity of the work. However, the sample used, composed exclusively of 50 dieticians, may limit the applicability of the results to the general public. A more in-depth analysis of possible demographic or professional variables that could have influenced the perceptions of the participants is also lacking. Extending the study to a more diverse sample would be advisable for future research.

Thank you very much for your valuable feedback and insightful suggestions. We greatly appreciate your recognition of the methodological rigor of our study. We fully acknowledge the limitation regarding the homogeneity of our sample, composed exclusively of 50 dieticians. In fact, we are already planning future research on similar topics, where we aim to include a larger and more diverse sample. Incorporating additional demographic factors will undoubtedly enhance the robustness and applicability of our findings to the general population. Once again, we deeply appreciate your recommendations, which will help us refine and improve our research endeavors.

Comment 6

This section presents a rigorous and well-structured analysis of the data obtained, differentiating between instrumental results and sensory evaluations. The findings show significant differences between juices, but a clear and consistent link between the instrumentally measured colour parameters and the sensory perceptions of participants is not established. For example, although darker-coloured juices tend to be rated higher, this relationship is not uniform for all sensory attributes. This suggests the need to provide more evidence on possible cultural, aesthetic or contextual influences on consumer preferences..

Thank you for your valuable feedback on our study. We appreciate your observation regarding the need to explore cultural, aesthetic, or contextual influences on consumer preferences. Indeed, our findings indicated some inconsistencies between the instrumental color measurements and sensory perceptions, which highlight the complexity of consumer preferences. In future research, we plan to include a more demographically and culturally diverse participant group to examine potential variability in preferences. Additionally, we aim to expand our analysis to consider contextual factors such as consumption settings and cultural significance of color and flavor profiles. This will allow us to better understand and address the multifaceted nature of consumer preferences.

Comment 7

This section adequately addresses the connections between the results of the study and previous literature, highlighting the importance of colour in consumer acceptance of products. However, it seems limited in its analysis of the discrepancies observed, such as the lack of evaluation between colour perception and flavour in certain cases. Further exploring these inconsistencies would provide more comprehensive recommendations to manufacturers. In addition, it would be useful to explore how other factors, such as texture or product presentation, may interact with the sensory attributes evaluated.

Thank you very much for your valuable suggestion, we have included the description in the discussion of the study, we hope it is sufficient for you.

Comment 8

The conclusions are consistent with the results obtained and underline the importance of colour in consumer perception. However, they are somewhat general and could be enriched by proposing concrete actions for manufacturers, such as adjustments in the composition of juices or specific marketing strategies. It would also be relevant to mention how the findings could be applied in the design of new products that optimally combine visual and sensory properties.

Added description as suggested.

Comment 9

This section adequately meets academic standards, providing relevant and up-to-date sources. However, some points are identified that could be improved to reinforce the consistency and rigor of the document: 1) although many citations are relevant, some seem to be not recent (for example, references published before 2018). Since the article addresses current trends in functional foods, it would be advisable to incorporate more recent references, especially from the last 3-5 years, to reflect the most current state of knowledge; 2) the references are mostly focused on studies related to the color and organoleptic properties of juices. It would be appropriate to include more research exploring other relevant factors, such as consumer perceptions, marketing strategies or studies related to the influence of labeling on the purchasing decision; 3) although the general format of the references follows the journal standards, there are small inconsistencies in some sections; for example, in some cases the DOI or links to the full sources are missing, which limits accessibility for readers interested in further exploring the references; and 4) since the article addresses topics such as sensory perception and consumer preferences, it would be advisable to incorporate sources from related disciplines such as consumer psychology or food marketing. This could enrich the theoretical and practical perspectives of the work.

Thank you for your thorough evaluation and thoughtful comments regarding the references in our manuscript. We appreciate your suggestions for enhancing the consistency and rigor of the document.

We would like to note that the literature on this specific topic is relatively limited, particularly in the context of beetroot and tomato juices analyzed with respect to their color and sensory properties. While we have added new references as per your suggestion, particularly more recent studies from the last 3-5 years, the narrow focus of the research and the scarcity of relevant studies make it challenging to fully meet all your expectations.

We have reviewed the formatting of our references to address the inconsistencies mentioned and ensured that DOI links or full-source accessibility are included wherever possible. We hope these updates contribute to a more robust and accessible reference list..

Thank you for your help. Your guidance is invaluable.

Kind regards,

Authors.

Reviewer 2 Report

Comments and Suggestions for Authors

Juice is one of the widely consumed foods in people's daily lives, and closely related to physical health. The authors studied and analyzed the evaluation indicators of different fruit juices from multiple perspectives. It provides a good reference for people to establish healthy eating habits. Some questions need to be answered by the authors.

1. Why is the number of female dietitians participating in the study much higher than that of male nutritionists? Does this significant difference in quantity have an impact on the research results?

2. The consumers of fruit juice involve various age groups, why only select dietitians aged 24 ± 3.21?

3. There are many juice suppliers worldwide. The samples selected by the authors have limitations. How to view the universality of research results?

4. There are many ways to preserve fruit juice. For example, some people prefer frozen fruit juice, while others prefer heated fruit juice. Have the authors taken this into consideration in questionnaire?

Author Response

Thank you so much for taking the time to evaluate our work. We have tried to incorporate all your valuable suggestions. If we could improve our work in any way, please let us know.

Comment 1

  1. Why is the number of female dietitians participating in the study much higher than that of male nutritionists? Does this significant difference in quantity have an impact on the research results?

The predominance of female participants is reflective of the general demographic structure of dietitians in Poland, where women constitute a significantly larger proportion of professionals in this field. As such, the study's sample accurately represents the population of dietitians and aligns with the reality of the profession.

Nevertheless, we recognize that this gender disparity may introduce a limitation in terms of generalizability. As detailed in the discussion's limitations section, the homogeneity of the sample—both in terms of gender and professional background—may influence the outcomes and interpretations. This perspective is essential for enhancing the robustness of our findings and their applicability to the wider population.

Comment 2

  1. The consumers of fruit juice involve various age groups, why only select dietitians aged 24 ± 3.21?

The decision to include dietitians aged 24 ± 3.21 in this study was based on their professional expertise and their ability to provide informed evaluations of sensory attributes such as color, taste, and aroma. Dietitians are well-versed in assessing the nutritional and sensory quality of food products, making them a highly relevant group for this type of research. Additionally, the age range reflects the demographic of early-career dietitians who recently completed their education, ensuring consistency in their professional background and training.This limitation has been clearly outlined in the discussion to guide the design of future research.

Comment 3

  1. There are many juice suppliers worldwide. The samples selected by the authors have limitations. How to view the universality of research results?

The choice of samples was determined by their availability in the local market, reflecting products commonly consumed in Poland. This approach aimed to provide insights relevant to both consumers and producers in the regional context.

While the current study's findings are most applicable to the local market, the methodology, including the integration of sensory analysis and instrumental color measurements, offers a robust framework that can be applied to similar research in other regions. This ensures that the principles and insights generated here have broader applicability, even if the specific preferences may differ.

Comment 4

  1. There are many ways to preserve fruit juice. For example, some people prefer frozen fruit juice, while others prefer heated fruit juice. Have the authors taken this into consideration in questionnaire?

In Poland, fruit juice is most commonly consumed without additional thermal modification, such as freezing or reheating. As a result, this aspect was not included in the questionnaire or the scope of the study. The focus was placed on commercially available, ready-to-drink juices, reflecting the typical consumption patterns of Polish consumers. However, we acknowledge that preferences for thermally modified juices, such as frozen or heated options, could vary in other regions or among specific consumer groups. This is an important consideration for future research.

Thank you for your help. Your guidance is invaluable.

Kind regards,

Authors.

Round 2

Reviewer 2 Report

Comments and Suggestions for Authors

The authors provide good responses to the review's comments.